# Laboratory Diagnosis of Intrathecal Synthesis of Immunoglobulins: A Review about the Contribution of OCBs and K-index

**DOI:** 10.3390/ijms25105170

**Published:** 2024-05-09

**Authors:** Maria Morello, Simone Mastrogiovanni, Fabio Falcione, Vanessa Rossi, Sergio Bernardini, Stefania Casciani, Antonietta Viola, Marilina Reali, Massimo Pieri

**Affiliations:** 1Clinical Biochemistry Department of Laboratory Medicine, Division of Proteins, University Hospital (PTV), 00133 Rome, Italy; simone.mastrogiovanni.01@alumni.uniroma2.eu (S.M.); fabio.falcione@ptvonline.it (F.F.); vanessa.rossi.06@students.uniroma2.eu (V.R.); bernards@uniroma2.it (S.B.); stefania.casciani@ptvonline.it (S.C.); antonietta.viola@ptvonline.it (A.V.); marilina.reali@ptvonline.it (M.R.); massimo.pieri@uniroma2.it (M.P.); 2Clinical Pathology and Clinical Biochemistry, Graduate School, Faculty of Medicine, University of Tor Vergata, 00133 Rome, Italy; 3Department of Experimental Medicine, Faculty of Medicine, University of Tor Vergata, 00133 Rome, Italy

**Keywords:** neurodegeneration, multiple sclerosis diagnosis, free light chains, nephelometry and turbidimetry, oligoclonal bands, CSF indexes k-FLC, K-index

## Abstract

The diagnosis of MS relies on a combination of imaging, clinical examinations, and biological analyses, including blood and cerebrospinal fluid (CSF) assessments. G-Oligoclonal bands (OCBs) are considered a “gold standard” for MS diagnosis due to their high sensitivity and specificity. Recent advancements have involved the introduced of kappa free light chain (k-FLC) assay into cerebrospinal fluid (CSF) and serum (S), along with the albumin quotient, leading to the development of a novel biomarker known as the “K-index” or “k-FLC index”. The use of the K-index has been recommended to decrease costs, increase laboratory efficiency, and to skip potential subjective operator-dependent risk that could happen during the identification of OCBs profiles. This review aims to provide a comprehensive overview and analysis of recent scientific articles, focusing on updated methods for MS diagnosis with an emphasis on the utility of the K-index. Numerous studies indicate that the K-index demonstrates high sensitivity and specificity, often comparable to or surpassing the diagnostic accuracy of OCBs evaluation. The integration of the measure of the K-index with OCBs assessment emerges as a more precise method for MS diagnosis. This combined approach not only enhances diagnostic accuracy, but also offers a more efficient and cost-effective alternative.

## 1. Introduction

MS is a neurodegenerative disorder in which both genetic and environmental factors play a role in promoting an abnormal immune reaction against self-components of the central nervous system (CNS) [1,2]. This progressive disease is characterized by demyelination and gliosis in a complex network of physiological and immune interactions [2,3]. In MS, the clinical evaluation of symptoms and the imaging studies (MRI), together with the analysis of cerebrospinal fluid (CSF) allows the following: (i) a precise diagnosis of the disease, (ii) an accurate monitoring of disease progression, and (iii) where possible, the evaluation of the effects of therapeutic strategies. The analysis of CSF and the alteration of some specific proteins in terms of concentration and forms are crucial for the differential diagnosis of patients affected by MS [4]. From a genetic point of view, high-throughput technology, which measures the simultaneous expression of thousands of genes, has shown high utility for the diagnosis of MS in ruling out MS-mimicking diseases in the early stages of the diagnostic process. Many single-nucleotide polymorphisms (SNPs) involving thousands of patients have been the subject of extensive genome-wide association studies (GWAS), which have identified 110 SNPs associated with MS [5]. Gurevich et al. created a blood gene-expression-based diagnostic classification tool, where findings of gene expression of MS-susceptible loci genes with other highly differentially expressed genes (DEGs) between MS and NonMS patients can be recorded. Interestingly, the authors demonstrated that, at the early stage of the disease, this diagnostic classifier successfully distinguished between CIS patients who were converted into MS and no-MS patients, with a sensitivity of 74.0% and a specificity of 76.0% [5]. Recently, the first meta-analysis of transcriptome investigations in MS has been carried out, evidencing the sex-related molecular processes driving MS [6]. The diagnostic criteria ascertained for MS require evidence of a demyelinating lesion that spreads temporally (DIT: dissemination in time) and spatially (DIS: dissemination in space) in the CNS and are associated with at least one clinical episode of neurological dysfunction [7]. In patients with a typical clinically isolated syndrome (CIS) (one episode of neurological disfunction with features suggestive of MS) and clinical or MRI evidence of DIS, the identification of immunoglobulins Igs) through CSF-specific oligoclonal bands (OCBs) enables the diagnosis of MS [8]. In the latest revision of the MS diagnostic criteria (2017), the detection of oligoclonal IgG bands (OCBs) in the CSF plays an extremely important role, as it allows the criterion of DIS to be replaced [8,9].

In MS, immunoglobulins (IgGs) are a type of antibody protein secreted from B cells and produced in higher quantities within the CNS through a specific inflammatory process called “intrathecal synthesis” [10]. The main involved Igs are IgGs, but the determination of intrathecal IgM appears to be of prognostic significance and is associated with a more aggressive MS disease progression [11]. In MS, these IgGs are all similar, because they are produced by the same cells, and during immunoelectrophoretic analysis, they appear as one or a few homogeneous and narrow bands, known as oligoclonal bands (OCBs) [3,8]. Moreover, IgGs production can be calculated by means of Link’s index, or the IgG index, given by the ratio of IgG quotient (QIgG) to albumin quotient (Qalb), or by means of Reiber’s function [2,12], representing a quantitative parameter. More specifically, Link’s index formula is the following: IgG index = (CSF IgG/IgG serum)/(CSF albumin/albumin serum); it was included in earlier versions of the McDonald criteria, but its diagnostic sensitivity appears to be low [11,13,14]. In both serum and CSF, the isoelectric focusing technique (IEF) is employed to identify IgGs [15]. This technique utilizes pH gradient electrophoresis on a nitrocellulose membrane and specific IgG bands are revealed through immunoelectrophoresis. When specific IgG bands (referred to as oligoclonal bands or OCBs) are detected exclusively in CSF and not in the serum, it suggests a specific activation of immune cells exclusively within the CNS rather than a generalized reaction of the immune system. Laboratory descriptions are often categorize in five common classic patterns: “type 1” (no bands are detected in CSF and serum, and it is considered negative); “type 2” (no bands are detected in serum, only CSF displays oligoclonal IgG bands), and this pattern is indicative of specific intrathecal IgG synthesis; “type 3” (oligoclonal IgG bands in both CSF and serum with additional oligoclonal bands in CSF), which is indicative of intrathecal IgG synthesis; “type 4”, (identical oligoclonal bands are detected in both CSF and in the serum). The presence of these bands in “type 4” are indicative of a systemic immune reaction and not of an intrathecal IgG synthesis, due to oligoclonal bands that can passively move to CSF. And, finally, pattern “type 5” is characterized by a monoclonal band in CSF and in the serum sample; this profile is present in MS-negative patients with plasma cell disease, polyneuropathy, organomegaly, endocrinopathy, M-protein, skin changes (POEMS) syndrome, and monoclonal gammaglobulinemia (MGUS) [16,17]. Pattern types 3 and 4 should be interpreted cautiously and type 4 is often characteristic for Guillain–Barré syndrome [17,18].

The presence of OCBs (type 2 and/or type 3 profiles) in CSF supports the diagnostic certainty for MS in terms of sensitivity and specificity aligning with the McDonald criteria [13,19] and highlighting their pivotal role—they are detected in >90% of MS cases [11]. In recent decades, several studies have explored potential biomarkers for MS. In CSF, the recognition of intrathecal IgG synthesis (detectable by OCBs), in conjunction with a high value of IgG index, can contribute to correctly diagnosing MS [16]. As reported in the McDonald criteria, the presence of OCBs in CSF is considered the gold standard for the qualitative diagnosis of MS [8]. Nevertheless, their determination is time-consuming and subject to an “eye meter analysis”, which requires high expertise and limits the execution to specialized centers [20,21]. Interestingly, in the last few years, new biomarkers for intrathecal synthesis have been discovered since, besides the intact immunoglobulins, an excess of free kappa (KFLC) and lambda light chains (λFLC) has been produced from plasma B cells and accumulated in the CSF of MS patients [21,22]

## 2. The Role of KFLC in CSF: A Significative Fluid Biomarker

Several studies have been published in the last decades demonstrating that the determination of free κ and λ light chains could represent a new quantitative marker of intrathecal synthesis of immunoglobulins [21]. During the immune reaction, FLCs are the protein chains that are normally produced from plasma cells and activated B cells. During the synthesis of immunoglobulins, FLCs, if not able to bind to the heavy chains of immunoglobulins, are released from the cells and FLCs increase, mainly in MS patients [22,23]. The most frequent tests used in the clinical laboratory for the detection and measurement of FLCs, are based on nephelometric or/and turbidimetric principles. Nephelometry techniques can detect the light scattered by particles in solution by measuring their concentration; while turbidimetry measures the turbidity of a solution by measuring the light scattered or absorbed due to suspended particles. Both techniques are automated, easy to use, cost-effective, and can provide precise concentration information of FLCs [24]. After the introduction of specific immunoassays for nephelometric or turbidimetric free light chain quantitation, several studies have shown higher KFLC index values in clinically isolated syndromes (CISs) or MS compared to other neurological diseases, while the results for free lambda light chains were incompatible [22,25]. Furthermore, the k-isotype of FLC is characterized by a monomeric structure, which might affect the high increment of CSF in cases of chronic intrathecal inflammation in comparison with the dimeric constitution typical of the λ-isotype [26,27]. Nonetheless, both k- and λ-isotypes FLC have a longer half-life in CSF despite having only a few hours of half-life in serum [26]. As a result, according to a recent study, CSF kFLC concentration is higher in MS patients compared to the other groups such as those with immune-mediated CNS disorders (IND), peripheral inflammatory disorders (PIND), non-immune-mediated neurological disorders (NIND), and symptomatic controls (SC), while no differences were found in serum FLCs. On the other hand, CSF λFLC concentration is higher in immune-mediated CNS disorder (IND) patients compared to MS and the other groups [28]. Moreover, recent studies have suggested that FLCs in CSF is a reliable marker of intrathecal immune activation, especially the K-FLC, which is normalized for to albumin quotient [18,22]. More specifically, the CSF FLC value must be corrected for blood–brain barrier (BBB) function to obtain the FLC index by dividing the FLC quotient (CSF FLC/serum FLC) by the albumin quotient (CSF albumin/serum albumin) to minimize the influence of blood–brain barrier permeability and the presence of a monoclonal serum component [22]. Precisely, the K-index and/or λ-index value is obtained via the ratio of the concentrations of the CSF and serum FLC over the ratio of the concentration of albumin in CSF and serum [26,29,30] (Figure 1).
$$FLCk\;Index=\frac{CSF\;FLCk/FLCk\;serum}{CSF\;albumin/albumin\;serum}$$
$$FLC\lambda \;Index=\frac{CSF\;FLC\lambda /FLC\lambda \;serum}{CSF\;albumin/albumin\;serum}$$

In particular, the K-index, compared to an analysis of OCBs, offers several advantages: (i) eliminates the operator-dependent risks (that it could happen during evaluation of the presence of OCBs); (ii) the measure of K-index is acquired within a relatively short timeframe (20–60 min); and, most importantly, (iii) the K-index serves as a quantitative metric [31].

Moreover, considering that McDonald’s criteria suggest caution and reasonableness in using the positive IgG index and the quantification of FLCs is not officially a gold standard yet [8,31], the ordinary use of the K-index demonstrates higher diagnostic accuracy and sensitivity compared to other classic MS biomarkers [16,32,33].

## 3. Laboratory Challenges in Intrathecal Synthesis Detection

Even though OCB detection is the gold standard for intrathecal immunoglobulin production, this test seems more appropriate as a follow-up methodology in samples chosen based on the findings of a more sensitive test, because the absence of CSF OCBs does not rule out MS, especially in the early stages of the condition [34]. In fact, a meta-analysis indicates that OCB sensitivity appears to be somewhat modest (69%) in patients with CIS, making it challenging to diagnose MS during the first stages of the disease [34]. Considering that, in the last years, (i) the use of K-index has been recommended as a preliminary tool for detecting intrathecal synthesis (useful for differential diagnosis of MS and neuroinflammatory diseases) and (ii) the measure of K-index has been proposed to better understand the progression of the pathology [16], the cut-off value of K-index remains heterogenous, poorly definite, and lacks standardization. It is pivotal to highlight that the identification of a reliable KFLC index cut-off would facilitate its use in clinical practice [21].

Moreover, the kFLC index may be used as an early predictor to enhance prognosis, diagnosis, and therapeutic response monitoring as the value of the K-index might increase from the CIS to the MS phase [35]. In order to contextualize its use in conjunction with OCB identification and the IgG index, we analyzed several clinical papers highlighting the positivity for OCBs and/or type profile bands, the techniques used, the value of the K-index (cut-off), and its measure of sensitivity and specificity (Table 1). In Figure 1 and Figure 2 we described, respectively, the frequency (%) of the K-index cut-off and the technique used. Figure 1 illustrates the variability of the K-index cut-off values: 72% of papers reported a K-index range cut-off 2.4–7.83, used to discriminate MS-positive patients from controls; 21% of papers used a cut-off between 8.33 and 9.58. Finally, to discriminate MS patients from MS-negative patients, only a few studies (7%) measured a K-index cut-off of between 10.62 and 12.3. In Figure 2, the techniques used—Freelite and N-Latex—are shown.

## 4. K-index: Population Characteristics

Our investigations, reported in Table 1, encompassed several papers that were performed in sizable population of patients affected by MS (ranging from a minimum of 70 patients to over 1000 individuals). This diverse cohort included various nationality, different age ranges (from 33 to 54 years), and both sexes. As reported in Table 1, in most of the included articles, the K-index and OCBs were obtained considering patients of similar ethnicity, mainly Europeans and Australians. In this context, we did not observe any significant differences regarding the potential impact of ethnicity on K-index.

Notably, sex-based variations in K-index values were observed in women exhibiting a sensitivity and specificity of 90.4% and 100%, respectively, at a K-index of 12.5. In contrast, men displayed values around 11 with sensitivity and specificity at approximately 97.5% and 100%. These findings align with the general reports indicating a higher prevalence of MS in the female sex, highlighting the influence of age, sex, and disease severity on K-index values [29,33,58].

These observations are consistent with the current reports in which men are, on average, older than women at the time of first diagnosis for MS and are more likely to develop a progressively severe form of the disease. In fact, according to several authors that studied the epidemiology of MS, the female sex to male sex ratio is approximately three to one: a pattern observed in various autoimmune diseases [29,59]. These insights further corroborate findings by Levraut et al. in which they reported that the age, female sex, and severity of disease activity could influence K-index values [33]. In addition, the K-index value, which is dependent from albumin concentration, may vary based on sex and age, as male end elderly patients exhibit different concentrations of albumin [30,33].

## 5. Diagnostic Performance of OCBs and K-index in MS

According to McDonald’s criteria, the evaluation of OCBs is considered to be the gold standard for the differential diagnosis of MS [8]. Our review of the pertinent literature revealed that the OCB value (reported as positivity/or as type of OCBs) showed a minimum value of sensitivity and specificity respectively: 37% for sensitivity [7] and 78% for specificity [33]. In addition, the higher value of OCB sensitivity was 100% [51] and 100% for specificity [7,45,50]. Therefore, the average of sensitivity, respectively, was 86.4% for sensitivity and 87% for specificity. In contrast, the value of the IgG index does not exhibit higher levels of sensitivity and specificity compared to OCBs. These data reinforce the idea that the qualitative identification of OCBs, more so than the measure of the IgG index, shows superior diagnostic performance and that OCBs evaluation is crucial to discriminate MS patients from control patients or those affected by other neurological diseases [22,54,60]. Clinically isolated syndrome (CIS) is characterized by a single monofocal lesion or multifocal lesions at the level of the brainstem, optics nerve, spinal cord, and cerebral hemisphere. CIS represents the earlier clinical expression of MS, and it is often linked to young adults that could later develop MS. As shown in Table 1, some articles that report the measure of OCBs with precise values of sensitivity and specificity allow the identification of MS patients, CIS patients, and patients affected by other neurological diseases (OND) [25,37,41,43].

The use of the K-index represents a quantitative measure of the presence of an intrathecal immune response, and it is important valuable proof for the initial diagnosis of MS [16]. As shown in Table 1, the value of the K-index reported is heterogenous; in fact, most papers reported a value of cut-off between 2.4 to 12.3, and these differences in values could not only be due to different populations and the severity of disease but also to the different kinds of free light kits used. In Table 1, the methods used are also specified: Freelite and N-Latex (Behring, Simiens, Spapalus, Behring, and Optlite).

In terms of diagnostic performance, the K-index exhibited a sensitivity range from 52% to 96% and a specificity between 68% and 100%. The average value of sensitivity of the K-index was 87% and specificity 88%—very close data to the average of OCBs sensitivity (OCBs = 86%) and OCBs specificity (OCBs = 87%). The higher value of K-index sensitivity was 96% [43] and 100% for specificity [50,58]. According to the values highlighted and reported in this review, a reliable K-index value between 4.6 and 6.6 showed comparable diagnostic sensitivity and specificity to OCBs [43,48,52,54]. Generally, a kFLC index cut-off equal to 4.7, compared to the OCB and IgG index, showed greater sensitivity (92.9%) but lower specificity (77.5%) [28]. In addition, considering the advantages regarding the use of the K-index (rapid and automated technique) compared to OCBs, some authors have proposed the use of the K-index as a primary marker instead of the evaluation of OCBs to highlight the intrathecal synthesis of IgG [3,28,38,61].

Figure 1 illustrates the variability in K-index cut-off values: 72% papers reported a K-index cut-off range of 2.4–7.83, used to discriminate MS-positive patients from controls; 21% papers used a cut-off between 8.33 and 9.58. Finally, to discriminate MS patients from MS-negative patients, only a few studies (7%) measured a K-index cut-off between 10.62 and 12.3.

According to most authors, it would be more appropriate to consider the range of the K-index cut-off between 2.4 and 7.8 [32,43,44,48,50], but several problems need to be solved: (i) the lack of an acknowledged global FLC reference standard, (ii) the paucity of information regarding the stability of the CSF matrix, and (iii) the numerous variables that affect kFLC concentration, such as blood [62]. For this reason and given the wide range of cut-off values obtained, it is pivotal to highlight the good agreement in CSF kFLC values throughout all laboratories, despite using various assays and platforms [62]. In this context, Natali et al. (2022) elucidated that there are two certified platforms available for CSF FLC tests (Binding Site and Siemens) which are based on distinct reaction principles and assays (Freelite and N-Latex). In 2011, Siemens was introduced to the market as a nephelometer with a monoclonal antibody-based assay. Binding Site, the earliest platform, has been available since 2001 and employs a turbidimeter. This paper highlights that K-index values between the two certified platforms and, more specifically, between laboratories using the same instrument and assay demonstrated excellent agreement. This can be attributed to the fact that the K-index is a ratio of values obtained from the same instrument, which removes the bias that arises when comparing absolute values [55].

In Figure 2, we can observe that the most common method used to measure KFLC is the Freelite analyzer (Binding Site) (as reported in 63.33% of the total papers). A total of 40% used N-latex (Siemens) as their analysis method. Notably, the Freelite analyzer offers a minimum K-index cut-off value of 5.9 (with sensitivity and specificity of 93% and 95%, respectively) and it seems to represent a sufficient minimum value to discriminate MS-positive patients from controls (no MS). Finally, these data demonstrate that the K-index shows excellent diagnostic performance for MS [25,40].

## 6. Relationships between the Number of OCBs and the Value of the K-index in MS Severity and/or Progression

In the papers considered in this review, the characterization (profiles) of OCBs was not consistently specified; instead, the term “positive” was occasionally reported to indicate the presence of OCBs. Some reports [7,33,40,41,45,55,56] described the presence of two or more OCBs in CSF, which were undetectable in serum. In only two papers, more than four OCBs were identified in CSF. When two or more OCBs were detected in CSF, the measured K-index varied from 4.6 to 10.6 [32,37].

From Table 1, an association between k-FLC index values and the number of OCBs seems clear. In fact, in patients with patterns 2 and 3 and with >6 CSF-restricted bands exhibited higher kFLC index values compared to those measured in CSF patients with 2 or 3 CSF-restricted bands [42,45]. To identify and separate patients affected by MS and CIS (CIS = early stage of MS, but not all CIS patients convert to MS), only some studies reported precise values of K-index [40]. From all the articles analyzed in this review, it emerges that the evaluation of OCBs is important, and it is strongly recommended in cases where MS diagnosis is uncertain [42]. Although, the discriminatory ability of the K-index (with good clinical sensitivity and specificity) is extensively and well described in all papers that we considered in this review. Only a few papers provided a precise value of the K-index linked to the progression of MS (see Table 1), [47,51]. In Table 1, a high K-index outcome that exceeds the value of 10.61 has been associated with an increased probability of conversion from CIS to MS [40,42,47,55,63]. Moreover, by comparing OCBs with the K-index, some authors reported that the latter is more sensitive but less specific for diagnosing CIS/MS [63]. In summary, while the observations regarding the utility of the K-index in MS diagnosis was well established, the studies about the value of K-index, progression, and risk of severe outcomes in MS are encouraging but still limited, necessitating at the further observations.

## 7. Laboratory Procedures

In the laboratory, adherence to recent guidelines is crucial. Paired samples of serum (S) and CSF should arrive promptly in the laboratory and should be processed within an hour of collection (to minimize cellular degeneration processes). CSF should be centrifuged for 10 min at 500 rpm at room temperature. Biochemical examination is conducted on the centrifuged CSF supernatant focusing the measure of glucose, albumin, and IgG, which are considered to be basic analytes. Glucose and albumin levels are determined in paired samples of S and CSF, expressing the percentage ratio. The albumin ratio in paired S and CSF is indicative of blood–brain barrier permeability (barrier damage) [39,64]. Regarding the evaluation of OCBs, there is no consensus on reporting. Common criteria include the following five conditions: (i) absence of bands; (ii) presence of single band; (iii) presence of two bands; (iv) presence of some bands (n = 3–6); and (v) presence of numerous bands (n > 6). If not tested immediately, CSF may be held for short periods at 4–8 °C and for the long term and at −80 °C. To better understand how the potential variations in experimental procedures could influence the value of the K-index we described the pre-analytical characteristics that were reported: centrifugation time and temperature of storage. The storage temperature depends on the analytes that needs to be analyzed: immunoglobulins (Ig) can be measured many years after storage at −20 °C, while other proteins, if not analyzed within 2 months, require a storage temperature of −80 °C. To ensure the stability of all markers, a storage temperature of −80 °C is recommended [53]. The detailed laboratory procedures reported in Table 1 reveal variations in storage methods: some authors stored the centrifugated cerebrospinal fluid samples at −80 °C [44,51,52,53,56] and others used samples that were centrifuged and stored at −20 °C until the moment of analysis [28,36,37,38]. Two papers described that the samples were immediately processed after collection [50,54], while one article reported that analyzed samples were processed two hours after collection and stored at 4 °C [57]. As shown in Table 1, despite these differences, no clear connections were observed between the reported K-index cut-off values and sample preservation methods.

## 8. Genetic Studies as Future Prospective

Recently, next-generation sequencing (NGS) studies have contributed to identifying the presence of a genetic influence on the OCB phenotype [65]. The association of MS to genes in the major histocompatibility complex (MHC) was established early [65], and carriers of the HLA-DRB1*15∶01 allele have more than three times increased risk for the disease [66]. In SNPs selected for replication, combined analyses showed genome-wide, significant association for two SNPs in the HLA complex, correlating with the HLA-DRB1*15 and the HLA-DRB1*04 alleles, respectively. In HLA-DRB1 analyses, HLA-DRB1*15∶01 was a stronger risk factor for OCB-positive than OCB-negative MS, whereas HLA-DRB1*04∶04 was associated with an increased risk of OCB-negative MS and a reduced risk of OCB-positive MS. Protective effects of HLA-DRB1*01∶01 and HLA-DRB1*07∶01 were detected [67,68]. Hence, some authors speculate that the HLA-DRB1*15∶01 allele possibly interacts with environmental agents in areas where this allele is common, making a population more prone to OCB-positive MS. The HLA-DRB1*04 allele may, on the other hand, confer a risk of OCB-negative MS via different immuno-genetic mechanisms [69]. In fact, alleles of the highly polymorphous HLA class II *DRB1* gene appear to be the strongest genetic determinant for MS and may influence both predisposition and resistance to the disease [70]. The distinct autoantigenic peptides presented by predisposing alleles have been identified. For instance, DRB1*15:01 binds peptides from the myelin basic protein (MBP), MBP_85−99_ peptide [67], while DRB5*01:01 presents the MBP_86−105_ peptide [71] and DRB1*04:01 can display the MBP_111−129_ peptide [72]. These findings define disease-associated peptide-HLA ligands recognizable by T-cells [73]. In addition, the association between exposure to household chemicals, pest control, and chemical-derived insecticides has been shown to be associated with the risk of developing MS, especially in pediatric onset [74,75]. In future, in order to obtain a specific diagnosis for MS, in advanced laboratories, it could be interesting to explore the association between genetic markers and OCBs.

## 9. Conclusions

MS is a complex interplay between internal and external factors, inflammatory processes, and neurodegenerative changes, which, chronically and progressively, potentially involve the entire central nervous system [10]. Epidemiological studies have revealed that genes are responsible for the highest specific weight within this disease. The major histocompatibility complex (MHC) has the greatest influence on the risk of developing MS, with its myriad of allelic variants. Besides genetic factors, environmental factors, such as viral infections, exposure to certain toxins, and vitamin D deficiency, may also contribute to the development of MS [76,77,78,79]. By means of genome-wide association (GWAS) studies, it has been shown that multiple variants play a key role in disease susceptibility [80].

In summary, we can conclude the following for MS diagnosis: (i) Comparing the main common markers used in MS, both the values of the K-index and OCBs detection exhibit excellent diagnostic performance (surpassing IgG index and λ -index). In fact, the majority of papers suggest that the K-index and OCBs show similar diagnostic accuracy in terms of sensitivity and specificity. (ii) The best minimum cut-off of the K-index indicated in the majority of papers falls within the range from 2.4 to 7.8 (Figure 2). (iii) The use of the Freelite method is advantageous for obtaining a minimum measure of the K-index (value 5.9) and offers a high value of sensitivity and specificity (Figure 3). In conclusion, from the data obtained from this review, it emerges that, in a laboratory specialized for MS diagnosis, the complementary association of the K-index (quantitative measure) with OCBs (qualitative evaluation) enhances diagnostic accuracy, thereby aiding clinician in achieving a better quality of diagnosis and prognosis, predicting, in some cases, disease severity and progression.

## Figures and Tables

**Figure 1 ijms-25-05170-f001:**
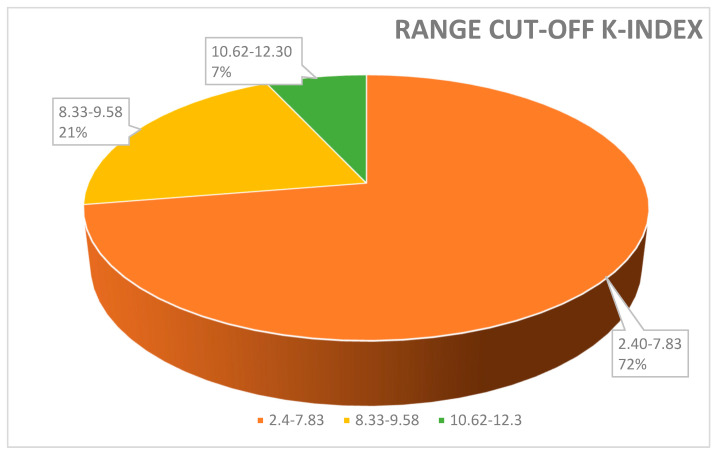
Range cut-off K-index used to discriminate MS-positive patients from controls. A total of 72% of papers reported a value of K-index cut-off between 2.4 and 7.83; 21% of papers used a cut-off between 8.33 and 9.58. Only a few studies (7%) measured a cut-off K-index range between 10.62 and 12.3%.

**Figure 2 ijms-25-05170-f002:**
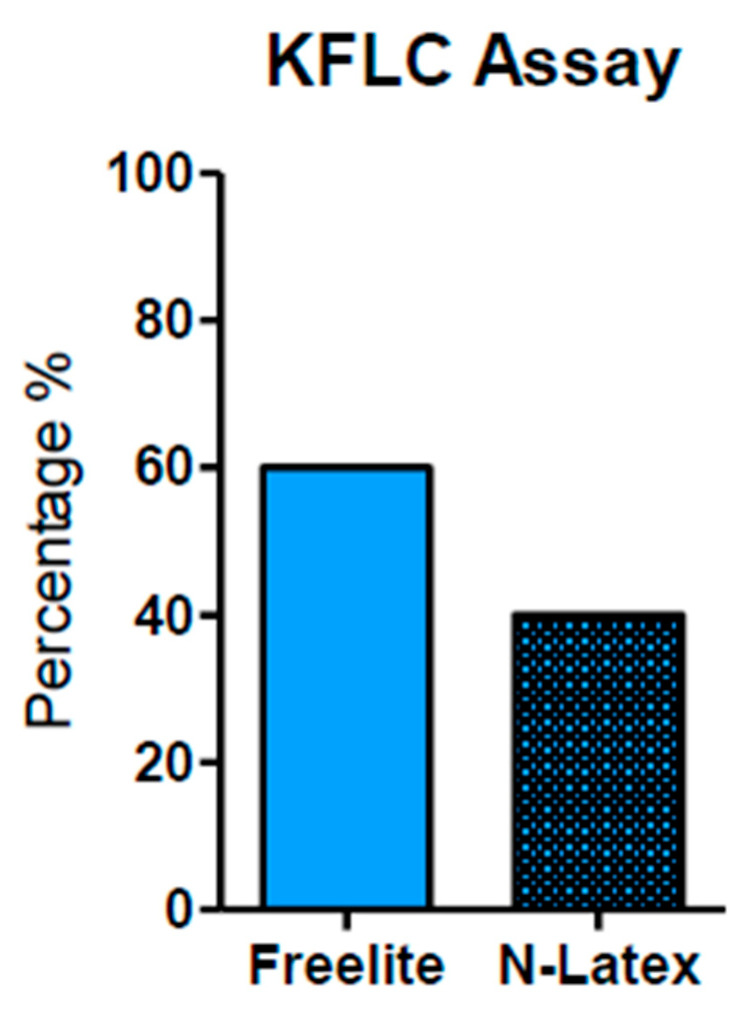
Techniques used for KFLC measurement. The Freelite analyzer method was used and reported in 63.33% of papers. The N-Latex method was reported in 40%. The Freelite analyzer offers a minimum K-index cut-off value of 5.9 (with sensitivity and specificity of 93% and 95%, respectively).

**Figure 3 ijms-25-05170-f003:**
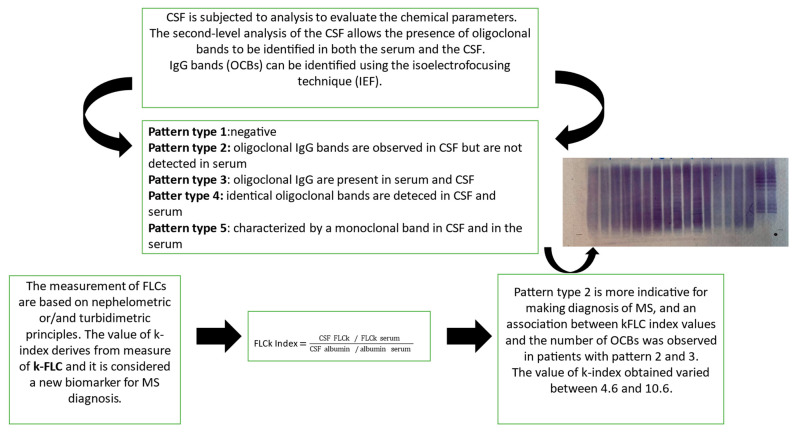
MS diagnosis: the common flowchart used in a laboratory. Both the K-index and isoelectrofocusing technique (IEF) are performed to characterize/confirm the presence of OCBs.

**Table 1 ijms-25-05170-t001:** Data about ethnicity, age, sex, and positivity for OCBs and type profile bands, the kind of techniques used, the value of the K-index (cut-off), and diagnostic measure (sensitivity and specificity of several CSF indexes).

Clinical Sensitivity and Specificity of Metrics in MS (Relevant Papers Published in Last Five–Six Years)													
Author [Ref.] [Publication]	Patients/Disease	Sex	Age (Medium)	Method	Metrics	OCB Profile	Cut-Off	Sensitivity %	Specificity %	Analyzer	KFLC Assay	Results	Experimental Procedures
Presslauer et al. [22] PMID: 18685917	70 patients (41 MS and 29 CISMS)	not specified	not specified	IEF	OCB	not specified	positive	91%	92%	BEHRING	FREELITE	The KFLC index was identified as a quantitative diagnostic parameter that is most sensitive and specific for MS. The specificity of the KFLC index for the MS group (0.86) was lower than that of the OCB group (0.92) but distinctly higher compared to the IgG index (0.77).	not specified
				nephelometry	IgG index		>0.6	80%	77%				
				nephelometry	MS KFLC index		≥5.9	96%	86%				
Presslauer et al. [25] PMID: 26199348	180 patients									BEHRING	FREELITE	The K-index was demonstrated to have diagnostic value in CIS/MS patients, highlighting that it is available as a simpler and evaluator-independent alternative to OCB detection.	not specified
	60 patients MS	47F, 13M	33 ± 11	IEF	OCB	not specified	positive	93.30%					
	60 patients CIS	45F, 15M	39 ± 14	IEF	OCB		positive	71.70%					
	60 patients OND	31F, 29M	48 ± 19	IEF	OCB		positive	1.70%					
	60 patients MS			nephelometry	IgG index		>0.6	80%					
							>0.7	73.30%					
	60 patients CIS			nephelometry	IgG index		>0.6	53.30%					
							>0.7	43.30%					
	60 patients OND			nephelometry	IgG index		>0.6	3.30%					
							>0.7	1.70%					
	60 patients MS			nephelometry	MS KFLC index		≥5.9	93%	95%				
	60 patients CIS			nephelometry	CIS KFLC index		≥5.9	78.30%	95%				
	60 patients OND			nephelometry	OND KFLC index		≥5.9	5%	95%				
Passerini et al. [36], PMID: 28116160	100 patients (34 MS, 22 CIS, and 44 C including 23 ID and 21 NO-ID)	38F, 18M	not specified	IEF	OCB	not specified	positive	71%	86%	BEHRING	N LATEX	Nephelometric tests for the KFLC index in CSF reliably detected intrathecal immunoglobulin synthesis and identified MS patients.	CSF and S samples were centrifuged for 10 min at 800 rpm and 10 min at 3000 rpm, respectively, and were stored at −20 °C until analysis.
				nephelometry	KFLC index		>2.4	89.30%	77.30%				
Christiansen, M. et al. [32], PMID: 30055097	230 patients (96 MS, 37 CIS, 31 OND, and 66 SC)	174F, 56M	40.5	IEF	OCB	≥4 liquor bands vs. serum	positive	82.30%	93.80%	SPAPLUS	FREELITE	The CSF-lambda and lambda-index performed inferior to the CSF-kappa and kappa index. The CSF-kappa and kappa index represent automated, rapid, and low-cost alternatives to OCB. Sensitivity and specificity for the diagnostic tests at different cut-offs were derived from the ROC curves. For quantitative tests, different cut-offs were reported to maximize both sensitivity and specificity and to allow for comparison.	not specified
	96 MS	24M, 72F	40	turbidimetry	IgG Index		>0.59	77.10%	93.80%				
				nephelometry	CSF KFLC		>0.42 mg/L	93.80%	85.60%				
				nephelometry	MS KFLC index		>5.2	93.80%	81.40%				
				nephelometry	MS KFLC index		>5.9	92.70%	86.60%				
	37 CIS	11M, 26F	39	IEF	OCB	≥4 liquor bands vs. serum	positive	82.30%	93.80%				
				turbidimetry	IgG index		>0.48	81.10%	48.50%				
				turbidimetry	IgG index		>0.59	59.50%	93.80%				
				nephelometry	CSF KFLC		>0.42 mg/L	70.30%	85.60%				
				nephelometry	CSF KFLC		>0.82 mg/L	67.60%	88.70%				
				nephelometry	CIS KFLC index		>5.2	70.30%	81.40%				
				nephelometry	CIS KFLC index		>5.9	70.30%	86.60%				
Gurtner, K.M. et al. [37] PMID: 29455184	325 patients (67 DD, 53 A, 50 non-inflammatory, 38 inflammatory, 31 others, 28 degenerative, 24 peripheral, 13 infection, 11 cancer, and 10 NMO/MOG)	174F, 151M	54	IEF	OCB	≥4 liquor bands vs. serum	positive	86.60%	89.60%	BEHRING	FREELITE	CSF KFLC demonstrated comparable performance to OCBs along with increased sensitivity for demyelinating disease. Replacing OCB with CSF KFLC is challenging. The measure of CSF KFLC can reduce cost and maintain sensitivity and specificity to support MS diagnosis.	Analyte stability was evaluated at five different time points over a 28-day period at room temperature, when refrigerated (4–8 °C), and when frozen (−20 °C), as well as during three freeze–thaw cycles. Analytical sensitivity studies included determination of the limit of quantification (LOQ) and analytical measurement range.
				IEF	OCB	≥3 liquor bands vs. serum	positive	88.10%	88.20%				
				IEF	OCB	2 liquor bands vs. serum	positive	94.00%	84.00%				
				nephelometry	IgG index		≥0.611	74.60%	80.00%				
				nephelometry	CSF KFLC		0.0611 mg/L	92.50%	86.10%				
				nephelometry	CSF KFLC		0.0875 mg/L	85.10%	89.60%				
				nephelometry	DD KFLC index		≥8.87	88.10%	88.70%				
Valencia-Vera, E. et al. [38] PMID: 29087953	123 patients (37 MS and 86 other pathologies in CNS)	not specified	not specified	IEF	OCB	not specified	positive	89.19%	81.18%	BEHRING	FREELITE	The KFLC index, rapid and automated, did not demonstrate sensitivity and specificity that was at least comparable to OCBs; however, it could be used as a screening tool. The IgG index and KFLC index cut-off values were estimated to distinguish MS from other neurological pathologies. The best values were 0.6 and 2.91, respectively. The KFLC index showed lower sensitivity (83.78%) than OCB (89.19%). A KFLC index above 8.33 is very suggestive of MS or another intrathecal synthesis disease.	CFSs were frozen at −20 °C until the KFLC analysis was performed.
				nephelometry	IgG index		>0.6	56.76%	94.19%				
				nephelometry	IgG index		>0.65	40.54%	94.12%				
				nephelometry	IgG index		>0.7	32.43%	94.12%				
				nephelometry	MS KFLC index		≥2.91	83.78%	85.80%				
				nephelometry	MS KFLC index		≥5.9	76%	91%				
				nephelometry	MS KFLC index		≥8.33	70.20%	95.60%				
Vasilj et al. [39] PMID: 30278299	151 patients (101 CIS/MS and 50 OND)	58F, 93M	40	IEF	OCB	≥2 liquor bands vs. serum	not specified	not specified	not specified	SIEMENS	FREELITE	The KFLC index showed diagnostic value but was not more specific and more sensitive than OCB. This method could be used as a screening tool, while OCB could only be used in uninterpreted cases. The KFLC index was high in patients that converted to MS, contributing to early diagnoses. ROC curve comparisons and comparisons of median KFLC parameters were used to find optimal cut-offs with regard to CIS diagnosis and conversion to MS.	An aliquot of S and CSF was stored at +4 °C until day 7, and another was stored at −20 °C until day 30.
				Nephelometry	MS KFLC index		>8.82	71.30%	98.00%				
Menendez-Valladares, P. et al. [40] PMID: 30408586	334 patients, 100 CIS	75F, 25M	36.51	IEF	OCB	≥2 liquor bands vs. serum	not specified	not specified	not specified	SIEMENS	FREELITE	The K-index showed high sensitivity and specificity and it value helps in MS diagnoses. A high K-index value increases the probability of conversion of CIS to MS.	not specified
				nephelometry	MS KFLC index		10.62	91%	89%				
Senel et al. [41] PMID: 30984199	1224 patients (75 MS, 1149 C including 36 AI-CNS-D, 5CIS 13 CIDP, 13 GBS, 29 CNS-D 7 TNC, 38 P-CNS-I, 5 ME, 14 PP-PNP, and 989 NIND)	629F, 595M	55.54	IEF	OCB	≥2 liquor bands vs. serum	positive	94.70%	93.30%	BEHRING	N LATEX	For MS, intrathecal KFLC and OCB showed the highest diagnostic sensitivity and lower specificity. Therefore, CSF KFLC might not replace OCB, but, as a quantitative parameter, it could support diagnosis in MS.	not specified
				turbidimetry	IgG index		>0.7	62.70%	98.30%				
				nephelometry	MS KFLC index		>9.58	92%	97%				
Crespi et al. [42] PMID: 30987052	385 patients (127 MS, 258 no-MS, 117 ID, and 141 NID)	not specified	48 ± 18	IEF	OCB	profile types 2/3	positive	0.97	0.83	BEHRING	N LATEX	The K-index showed higher sensitivity in predicting OCB and diagnosing MS.	not specified
					MS K-index		>5	96.5%	89.80%				
Emersic et al. [43] PMID: 30529605	130 patients (78 RR-MS, 1 PP-MS, 1 SP-MS, and 50 controls (NIND))	82F, 48M	not specified	IEF	OCB	≥2 liquor bands vs. serum	positive	91.30%	98%	BEHRING	N LATEX	Normal KFLC parameters allow safe exclusion of intrathecal inflammation. The K-index could be useful in specialized MS centers as an additional pre-test (screening test).	not specified
					IgG index		>0.6	60.30%	100%				
					MS KFLC index		>5.30	96%	96%				
Schwenkenbecher et al. [44] PMID: 31744096	100 patients MS	73F, 27M	32	IEF	OCB	not specified	positive	0.99	not specified	SIEMENS	N LATEX	In MS patients, Reiber’s KFLC diagram showed excellent diagnostic performance in detecting diagnostic performance.	Paired CSF and S samples were collected as part of a routine diagnostic work-up. The samples were collected and stored at −80 C° within one day after sampling.
Reiber et al. [45] PMID: 31351929	351 patients: (95 MS and 256 controls)	not specified	not specified	IEF	OCB	≥2 liquor bands vs. serum	positive	94%	100%	SIEMENS	N LATEX	This study primarily focused on the development of an empirically and theoretically correct reference range (>5). It could be used in pathophysiological and clinical applications.	not specified
				nephelometry	MS KFLC index		>5.00	93%	94%				
Saez et al. [46] PMID: 30386877	119 patients (36 CIS, 41 RR-MS, 42 C including 35 headache complaints, and 7 NO-ID	21F, 15M:	not specified	IEF	OCB	not specified	positive	93%	90%	SPAPLUS	FREELITE	KFLCs have high sensitivity and specificity for the diagnosis of MS. FLC concentration at CIS diagnosis was not significantly higher in CIS converters.	After routine diagnostic work-up, excess volumes of CSF/S pairs were stored immediately at −80 °C util further analyses.
				turbidimetry	MS KFLC index		>7.10	95%	97%				
Gaetani et al. [47] PMID: 31743879	170 patients (64 RIS, CIS, RR-MS, PMS and 106 controls 3 RIS, 23 CIS, 34 RR-MS, and 4 PMS)	48F, 12M	40 ± 12.3	IEF	OCB	≥2 liquor bands vs. serum	positive	83%	92%	BEHRING	FREELITE	A K-index > 7.83 was more sensitive but less specific than OCB in discriminating MS patients from controls. Additionally, a κ-index > 10.61 performed better than OCB in the discrimination between MS and controls with inflammatory neurological diseases < 0.001. In clinically isolated syndrome (CIS) patients, a K-index >10.61 significantly predicted time to conversion to MS (*p* = 0.020). The K-index might be a valid alternative to OCB as a biomarker for MS and might also be a prognostic marker for CIS.	not specified
				nephelometry	MS KFLC index		>7.83	89%	81%				
Cavalla et al. [48] PMID: 31837636	373 patients (140 MS, 233 C whose 8 NMOSD, 32 OICDs, 37 PNS, and 156 NI-CNSD)	26.8% M, 73.2% F	46.3 ± 18.9, 40.5 ± 1.3	IEF	OCB	not specified	positive	86%	89%	BEHRING	N LATEX	The KFLC index was proposed as a marker for discrimination between patients with MS and other CNS disorders. The authors showed an increase in the value of the KFLC index in MS patients compared with other CNS disorders. KFLC index values >100 were found exclusively in MS patients.	not specified
				nephelometry	MS KFLC index		>6.15	89%	81%				
Leurs et al. [49] PMID: 31066634	745 patients (526 CIS/MS; 219 C including 76 NINDC, 67 INDC, 49 SC, and 27 HC)	170F, 114M	38 ± 10	IEF	OCB	not specified	positive	82%	92%	OPTILITE	FREELITE	Compared with OCB, the KFLC index is more sensitive but less specific for diagnosing CIS/MS. Lacking an elevated KFLC index is more powerful for excluding MS compared with OCB, but the latter is more important for ruling in a diagnosis of CIS/MS.	CSF samples were immediately centrifuged and stored in polypropylene tubes (within 2 h at −80 °C). OCB evaluation was performed via isoelectric focusing (on agarose or polyacrylamide gels), followed by IEF.
				turbidimetry	MS KFLC index		>6.6	88%	83%				
Duell et al. [50] PMID: 32527760	122 patients (62 RR-MS and 60 C)	not specified	not specified	IEF	OCB	not specified	positive	87%	100%	BEHRING	N LATEX	The KFLC parameters evaluated had excellent accuracy. The evaluated KFLC index had equal or higher sensitivity than OCB in discriminating MS patients from the control group without considerable loss of specificity.	For all analyses, samples derived from the same sampling (CSF and S) were used, i.e., when sampling (CSF and S), several aliquots were collected fresh and sent simultaneously to the laboratory for analysis.
				nephelometry	CSF KFLC		>0.47 mg/L	89%	98%				
				nephelometry	MS KFLC index		>7.15	90%	100%				
Gudowska-Sawczuk et al. [51] PMID: 32471086	76 patients (34 RR-MS and 42 C)	27F, 7M, 29F, 13M	35; 47	IEF	OCB	not specified	positive	100%	90%	OPTILITE	FREELITE	This study provided novel information about the diagnostic significance of four markers combined in the kIgG index. The kFLC index has high sensitivity and probably would avoid OCBs determination in most patients with suspected MS.	CSF samples were collected in polypropylene tubes, centrifuged, aliquoted, and frozen at −80 °C until assayed. Venous blood samples were collected and centrifuged to separate S. S samples were aliquoted and frozen at −80 °C until assay.
				Turbidimetry	MS KFLC index		>9.42	94%	68%				
Ferraro et al. [7] PMID: 31710409	445 patients (146 patients MS and 299 C (OCB-negative NO-MS))	105F, 41M, No MS: 184F, 115M	40, No MS:43 ± 15	IEF	OCB	≥2 liquor bands vs. serum		37%	100%	OPTILITE	FREELITE	A kappa index ≥5.8 was measured in 25% of OCB-negative MS (23/92) and in 98% of OCB-positive patients with MS.	not specified
				turbidimetry	MS KFLC index		>5.8	52%	94%				
Ferraro et al. [52] PMID: 33096861	542 patients (84 patients MS and 458 controls (NO-MS))	CTRL (222 F, 236M) MS (54 F, 30 M)	CTRL (57± 20) MS (38 ± 14)	IEF	OCB	not specified	positive	85%	89%	OPTILITE	FREELITE	The CSF OCB and kappa index have a comparable diagnostic accuracy in the prediction of MS or for CNSID. The K-index has a slightly higher sensitivity and lower specificity than CSF OCB. The OCB and K-index supply clinicians with useful, complementary information. The optimal kappa index cut-off for a MS diagnosis using the Youden Index was 6.2.	CSF and S were centrifuged at 3000 rpm (rotations per minute) for 10 min and stored in cryogenic tubes at −80 °C within two hours.
				turbidimetry	MS KFLC index		>6.20	89%	84%				
Sanz Diaz et al. [53] PMID: 34456842	242 patients (35 MS, 1 CIS, 2 RIS, 7 inconclusive, and 197 other diagnosis)	not specified	41	IEF	OCB	≥2 liquor bands vs. serum	positive	88.89%	90.86%	OPTILITE	FREELITE	The K-index in this work showed higher sensitivity and slightly lower specificity than the gold standard for immunoglobulin detection (OCB). Patients with MS had significantly higher K-index levels than patients without a diagnosis of a MS.	Cerebrospinal fluid and S samples were kept frozen at −80 °C until analysis.
				turbidimetry	MS KFLC index		>6.6	93.30%	87.31%				
Rosenstein et al. [54] PMID: 36742305	324 patients (23 SP-MS, 20 CIS/RIS, 161 RR-MS, 19 PP-MS, 74 symptomatic controls, and 27 ONID)	123F, 101 M	41 ± 13	IEF	OCB	≥2 liquor bands vs. serum	positive	88%	89%	SIEMENS	N LATEX	CIS, RIS, and MS had a significantly higher KFLC index than the controls. The KFLC index had a sensitivity of 0.93 and specificity of 0.87 to discriminate CIS/RIS/MS from ONID and SC. The KFLC intrathecal index (IF) had similar accuracy in detecting MS. Disease-modifying treatment (DMT) did not affect the level of the KFLC index.	Paired CSF and S samples were obtained during a routine diagnostic work-up and analyzed consecutively (fresh, no frozen).
				Nephelometry	IgG index		>0.53	85%	77%				
				nephelometry	MS KFLC index		>4.6	93%	87%				
Berek et al. [55] PMID: 34049994	88 patients (syndrome monofocal)	60F, 28M	33 ± 10	IEF	OCB	≥2 liquor bands vs. serum	positive	90%	not specified	BEHRING	N LATEX	A high K-index predicts early MS disease activity.	CSF freshly processed
				nephelometry	MS KFLC index		>6.60	86%	not specified				
Bernardi et al. [27] NO PMID	406 patients (171 MS, 235 controls, 154 NIND, 48 IND, and 33 PIND)	170F, 236M	49.8	IEF	OCB	≥2 liquor bands vs. serum	positive	84%	91%	OPTILITE	FREELITE	A KFLC index ≥ 6.4 is comparable to OCB for MS diagnosis. In all, 12/27 (44.4%) MS patients with negative OCB had a kFLC index ≥ 6.4.	not specified
				turbidimetry	MS KFLC index		>6.40	84%	89%				
Levraut et al. [33] PMID: 36376096	675 patients (591 RR-MS, 14 SP-MS, and 69 PP-MS)	481F, 194M	37	IEF	OCB performance in separating MS/CIS from the control.	≥2 liquor bands vs. serum	positive	81.99%	90.16%	OPTILITE	FREELITE	KFLC biomarkers are effective tools for separating MS patients from controls (even when compared with other patients with autoimmune disorders of the CNS). In this cohort, a KFLC index > 8.92 permits separation of MS from controls, with better sensitivity and the same specificity as OCB, and a KFLC index > 11.56 permits separation of MS form other CNS diseases.	not specified
					OCB performance in separating from other inflammatory CNS disorders	≥2 liquor bands vs. serum	positive	84.26%	78.09%				
				nephelometry/turbidimetry	CSF KFLC		>0.94 mg/L	86.73%	75.62%				
				nephelometry/turbidimetry	MS/CIS KFLC index		>8.92	88.24%	89.36%				
				nephelometry/turbidimetry	MS KFLC index		>11.56	87.56%	79.73%				
Monreal et al. [56] PMID: 37954589	371 patients (41 CIS and 330 MS)	260F, 111M	34.9	IEF	OCB	≥2 liquor bands vs. serum	positive	95.30%	100%	OPTILITE	FREELITE	The combination of the KFLC index and OCB can provide an accurate and easily reproducible method for MS diagnosis.	Paired S and CSF samples were obtained and stored at −80 °C until analyzed.
				turbidimetry	CIS KFLC index		>6.1	86.30%	93.90%				
					MS KFLC index		>6.6	85.60%	86.40%				
ZK Revendova et al. [57] PMID: 37980789	1751 patients (60 CIS/RIS, 168 INDC, 52 PINDC, 464 NINDC, 628 SC, 311 RR-MS, 62 PP-MS, and 6 SP-MS)	1117F, 634M	37	IEF	OCB	≥2 liquor bands vs. serum	positive	95.60%	86.90%	SPAPLUS	FREELITE	OCB shows higher sensitivity and, thus, remains the gold standard for MS diagnosis.	S and CSF samples were taken on the same day. CSF samples were centrifuged at 390× *g* for 10 min at room temperature. S samples were centrifuged at 2500× *g* for 6 min at 4 °C. For albumin, aliquots of IgG, KFLC, and OCB were stored at 4 °C for up to 1 week and for KFLC analysis for up to 3 weeks.
				Nephelometry	MS KFLC index		>8.93	92.50%	93.50%				
Michetti et al. [28] PMID: 37210840	122 patients (42 MS, 29 IND, 26 NIND, 18 PIND, and 7 SC)	56M, 66F	45.9									The authors reported an association between KFLC values and the number of bands in OCB-positive patients (after follow-up). KFLC discriminates between MS and other neurological diseases with lower sensitivity and specificity than OCB, respectively. It is recommended to use the KFLC index as a first-line marker followed by IEF.	Upon sample collection, the paired CSF fluid and S were centrifuged at 2000× *g* and 3500× *g* for 10 min, respectively, and stored in cryogenic tubes at −20 °C until further analysis.
	42 MS			IEF	OCB	≥2 liquor bands vs. serum	positive	85.70%	88.60%	OPTILITE	FREELITE		
	42 MS			turbidimetry	IgG index		≥0.7	73.80%	76.20%				
	42 MS			turbidimetry	MS KFLC index		≥4.7	92.90%	77.50%				
Pieri et al. [58] PMID: 30223231	228 patients tot	131F, 97M		nephelometry	MS KFLC index		12.3	93%	100%	SIEMENS	N LATEX	The cut-off distance of the K-index in woman and men was shown to be higher and lower, respectively. The authors suggested that it could be related to sex hormones.	not specified

A: autoimmune; AI-CNS-D: autoimmune CNS diseases; C: controls; CSF: fluid cerebrospinal; CIDP: chronic inflammatory demyelinating polyneuropathy; CIS: clinically isolated syndrome; CISSMS: clinically isolated syndrome suggestive of MS; CNS-I: CNS infection; DD: demyelinating disease; GBS: Guillain–Barré syndrome; HC: healthy controls; ID: disease inflammation; INDC: inflammatory neurological disease controls; ME: metabolic encephalopathy; MS: sclerosis multiple; NI-CNSD: non-immune-mediated CNS disorders; NIND: non-inflammatory neurological diseases; NMO/MOG: neuromyelitis optica/myelin oligodendrocyte glycoprotein; NMOSD: neuromyelitis optica spectrum disorder; OICDs: immune-mediated CNS disorders; OND: other neurological diseases; ONID: neuroinflammatory disorders; P-CNS-I: post-infectious CSF syndrome; PIND: peripheral inflammatory neurological diseases; PNS: peripheral nervous system; PMS: progressive multiple sclerosis; PP-MS: primary progressive—multiple sclerosis; PP-PNP: paraproteinemic neuropathy and/or neuropathy with monoclonal gammopathy of unknown significance; RR-MS: relapsing-remitting—multiple sclerosis; S: serum; SC: symptomatic controls; SP-MS: secondary-progressive—multiple sclerosis; TNC: CNS tumor; Tot: totals.

## Data Availability

All the date used in this manuscript are available on Pubmed.

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
