# Peer review of "Laboratory Diagnosis of Intrathecal Synthesis of Immunoglobulins: A Review about the Contribution of OCBs and K-index"

_ijms, 2024, doi:10.3390/ijms25105170_

Round 1

Reviewer 1 Report

Comments and Suggestions for Authors

The manuscript titled "Laboratory Diagnosis of Intrathecal Synthesis of Immunoglobulins: A Review of the Contribution of OCBs and K Index" by Maria Morello et al. offers a thorough review that consolidates the current understanding of the diagnostic value of oligoclonal bands (OCBs) and the kappa free light chains (k-FLC) index, specifically the "k-Index," within the framework of Multiple Sclerosis (MS) diagnosis. The authors provide a comprehensive examination of the dynamic field of MS diagnosis, focusing on the amalgamation of traditional and emerging biomarkers to improve diagnostic precision, laboratory efficiency, and cost-effectiveness.

Although the manuscript recognizes the variability in k-Index cutoff values across different studies, it would benefit from an in-depth exploration of the methodological disparities contributing to this variation. A discussion on the association of the disease with other biomarkers, such as single nucleotide polymorphisms (SNPs), genes, and references to literature like ktwas, mktwas (for transcriptome-wide association studies), etc., would provide a more holistic view of the diagnostic landscape.

The manuscript could also gain from a detailed discussion on the challenges and limitations inherent in diagnosing MS, especially during the initial stages or in cases with atypical presentations. This nuanced conversation could illuminate the complexities of early detection and diagnosis, providing valuable insights for clinicians.

While the manuscript ends with robust recommendations for employing the k-Index in conjunction with OCBs, there is room to expand on areas where further research is necessary. Future studies could aim to validate and refine the k-Index's diagnostic capability, offering clearer guidance on its application in clinical practice. Such elaboration would not only underscore the current gaps in knowledge but also highlight the potential directions for upcoming research in this field.

Comments on the Quality of English Language

there are opportunities to enhance clarity and readability further.

Author Response

The manuscript titled "Laboratory Diagnosis of Intrathecal Synthesis of Immunoglobulins: A Review of the Contribution of OCBs and K Index" by Maria Morello et al. offers a thorough review that consolidates the current understanding of the diagnostic value of oligoclonal bands (OCBs) and the kappa free light chains (k-FLC) index, specifically the "k-Index," within the framework of Multiple Sclerosis (MS) diagnosis. The authors provide a comprehensive examination of the dynamic field of MS diagnosis, focusing on the amalgamation of traditional and emerging biomarkers to improve diagnostic precision, laboratory efficiency, and cost-effectiveness. Although the manuscript recognizes the variability in k-Index cutoff values across different studies, it would benefit from an in-depth exploration of the methodological disparities contributing to this variation. A discussion on the association of the disease with other biomarkers, such as single nucleotide polymorphisms (SNPs), genes, and references to literature like ktwas, mktwas (for transcriptome-wide association studies), etc., would provide a more holistic view of the diagnostic landscape.

- Thank you for the appreciation and drawing our attention to this. A specific paper about the Inter-Laboratory concordance of serum kappa free Light chain measurements (Natali et al., 2022) shows a good correlation among diverse measurements achieved, despite the methodological disparities. These informations are comprised in the paragraph “Diagnostic Performance of OCBs and K-Index in MS” (between lines 259 -275)

- Thank you very much for the suggestion, we have included  new references in introduction about the importance of SNPs and the transcriprome-wide association studies (from line 41 to 65). In addition, we have written a new paragraph “Genetic studies as future prospective” from lines 340 to 367.

The manuscript could also gain from a detailed discussion on the challenges and limitations inherent in diagnosing MS, especially during the initial stages or in cases with atypical presentations. This nuanced conversation could illuminate the complexities of early detection and diagnosis, providing valuable insights for clinicians. While the manuscript ends with robust recommendations for employing the k-Index in conjunction with OCBs, there is room to expand on areas where further research is necessary. Future studies could aim to validate and refine the k-Index's diagnostic capability, offering clearer guidance on its application in clinical practice. Such elaboration would not only underscore the current gaps in in knowledge but also highlight the potential directions for upcoming research in this field

Thank you for the suggestion. We inserted a specific insight about the complex diagnosis at the first diagnostic stage of MS evidencing the complementary value of OCBs in conjunction with the k-index. Furthermore, the K-index has an increasing value depending on the stage of MS diseases that might help the clinicians to assert the definitive medical outcome (paragraph “5. Diagnostic performance of OCBs and K-index in MS” from line 218 to 236 and line 160 to 193.

Reviewer 2 Report

Comments and Suggestions for Authors The manuscript presented from Maria Morello et al., entitled "Laboratory Diagnosis of intrathecal synthesis of immunoglobulins: a review about the contribution of OCBs and K index" is a comprensive review. Comparing with other published studies could be considered original. However the authors must improve different critical point: - The authors must improve the number of reference in the text, there are several lines without reference for ex. from line 1 to 66 (the authors reported only 5 references), the same for following paragraphes - The authors must improve the quality of english (there are many sentences that do not make sense; see line55; line68-75; line 105-108; line114-121; line143-149 etc...) - The authors should discuss the limit of their experimental approach (if there are), and they should mention furture perspective.   -            Comments on the Quality of English Language       - The authors must improve the quality of english (there are many sentences that do not make sense; see line55; line68-75; line 105-108; line114-121; line143-149 etc...)            

Author Response

The manuscript presented from Maria Morello et al., entitled "Laboratory Diagnosis of intrathecal synthesis of immunoglobulins: a review about the contribution of OCBs and K index" is a comprensive review. Comparing with other published studies could be considered original. However, the authors must improve different critical point: -

-The authors must improve the number of references in the text, there are several lines without reference for ex. from line 1 to 66 (the authors reported only 5 references), the same for following paragraphs -The authors must improve the quality of English (there are many sentences that do not make sense; see line55; line68-75; line 105-108; line114-121; line143-149 etc...)

Thank you for drawing our attention to this, we have reviewed the English language, revisited the introduction (Lines 34-109), included references and reconsidered paragraphs to enhance readability and accuracy in presentation

The authors should discuss the limit of their experimental approach (if there are), and they should mention future perspective. 

Thank you for the suggestions, we  have provided a more detailed explanation  about the limit of experimental approach used (form lines 212 to 217, 237-243) and lines 259 to 275; we have integrated the review with a new paragraph about the future perspective: “Genetic studies as future prospective” (lines 341-365) and revisited conclusions (lines 368-389)

Reviewer 3 Report

Comments and Suggestions for Authors

This review is well conceived and cover all the results coming from a good amount of papers dealing with usefullness of K-index in diagnosing MS.

I have just one question: from the reviewed literature does emerge a possible role of K-index in achieving a diagnosis of MS in OCB- patients? 

Some minor style issues:

- line 210-213, the sentence is not clear, I think some verbs are missing, please reformulate

- throughout the paper please correct SM>MS, OBC>OCB.

Author Response

This review is well conceived and cover all the results coming from a good amount of papers dealing with usefulness of K-index in diagnosing MS. I have just one question: from the reviewed literature does emerge a possible role of K-index in achieving a diagnosis of MS in OCB- patients? 

Thank you for the appreciation and drawing our attention to this. From the analysis of data included in this review emerges that OCBS and K-index are important biomarkers for MS; in particular, the observations collected in this article suggest that  rather than the use of a single marker: OCBs or K-index, the combined measure of k-index value with qualitative evaluation of OCBs, enhances diagnostic accuracy and support the clinicians to better understand the progression of severity of MS ( paragraph 6“relationships between the number of OCBs and the value of K-Index in MS Severity and/or Progression,  lines from 283 to 308.

Some minor style issues: - line 210-213, the sentence is not clear, I think some verbs are missing, please reformulate. - throughout the paper, please correct SM>MS, OBC>OCB

Thank you so much for the advice, we have reviewed the English language and correct the abbreviations.

Round 2

Reviewer 2 Report

Comments and Suggestions for Authors

The authors satisfied all my concerns.